# Language of Driving for Autonomous Vehicles

**Krister Kalda** [1,*], **Simone-Luca Pizzagalli** [2], **Ralf-Martin Soe** [1], **Raivo Sell** [2] and **Mauro Bellone** [1]

1    Finest Centre for Smart Cities, Tallinn University of Technology, 19086 Tallinn, Estonia;
ralf-martin.soe@taltech.ee (R.-M.S.); mauro.bellone@taltech.ee (M.B.)

2    Department of Mechanical and Industrial Engineering, Tallinn University of Technology,
19086 Tallinn, Estonia; simone.pizzagalli@taltech.ee (S.-L.P.); raivo.sell@taltech.ee (R.S.)

\*    Correspondence: krister.kalda@taltech.ee; Tel.: +372-58835557

**Abstract:** Environmental awareness and technological advancements for self-driving cars are close to making autonomous vehicles (AV) a reality in everyday scenarios and a part of smart cities' transportation systems. The perception of safety and trust towards AVs of passengers and other agents in the urban scenario, being pedestrians, cyclists, scooter drivers or car drivers, is of primary importance and the theme of investigation of many research groups. Driver-to-driver communication channels as much as car-to-driver human–machine interfaces (HMI) are well established and part of normal training and experience. The situation is different when users must cope with driverless and autonomous vehicles, both as passengers and as agents sharing the same urban domain. This research focuses on the new challenges of connected driverless vehicles, investigating an emerging topic, namely the language of driving (LoD) between these machines and humans participating in traffic scenarios. This work presents the results of a field study conducted at Tallinn University Technology campus with the ISEAUTO autonomous driving shuttle, including interviews with 176 subjects communicating using LoD. Furthermore, this study combines expert focus group interviews to build a joint base of needs and requirements for AVs in public spaces. Based on previous studies and questionnaire results, we established the hypotheses that we can enhance physical survey results using experimental scenarios with VR/AR tools to allow the fast prototyping of different external and internal HMIs, facilitating the assessment of communication efficacy, evaluation of usability, and impact on the users. The aim is to point out how we can enhance AV design and LoD communications using XR tools. The scenarios were chosen to be inclusive and support the needs of different demographics while at the same time determining the limitations of surveys and real-world experimental scenarios in LoD testing and design for future pilots.

**Keywords:** AV shuttle; self-driving vehicle; language of driving; simulations; interaction

## 1. Introduction

Autonomous vehicles (AVs) are one of the dominant topics in engineering research, and a large number of private and academic organizations are investing resources to develop effective autonomous agents that will be populating our streets in the coming years. One major dilemma faced by autonomous cars is understanding the intentions of other road users and how to communicate with them [1]. Though localization, mapping, route planning, and control of AVs are widely studied, and the literature already offers working solutions in static environments, road users are non-static, complex, interactive agents having their own goals, utilities, and decision-making systems [2]; their study is an emerging research topic. Human–AV interaction (HAVI) with pedestrians must take these interactive agents into account in order to predict their actions and plan accordingly. The mode and intent of communication from the AVs to other road users are two fundamental requirements for the future of transportation systems. Clear messages and information are necessary to avoid misunderstandings between vehicles and pedestrians leading to unexpected behaviors, unnecessary vehicle yielding, or dangerous situations.

The current driving domain is dominated by human drivers and well-established communication channels based on human-to-human (eye contact, head positions, gestures) and machine-to-human (speed, direction, sound) cues [3]. This is supported by well-established and defined methods to communicate the vehicle status and intentions. This situation will slowly shift to a mixed agent domain, where autonomous vehicles will coexist together with human drivers. The fast development of these technologies foresees a future scenario where autonomous vehicles will dominate road traffic, and the presence of human-controlled agents will be minimal. Nevertheless, the presence of human agents or other road users in the traffic flow will probably endure. In this context, AVs should be able to communicate their status in a clear and understandable way, by making use of specific Human–Machine Interface (HMI) systems and interpreting human decisions by detecting pedestrians' movement cues and behavioral patterns [4].

HMI solutions taking into consideration the new Language of Driving (LoD) between machines and human agents need to be assessed and validated through user-based research studies. Furthermore, different road users' needs and abilities should be taken into consideration. Pedestrians and AV shuttle passengers might vary in different contexts or moments of the day. Scenarios like big hospital compounds, schools, or simply residential districts need to be considered with children, elderly people, and other fragile user groups being involved in testing and assessing HMI solutions for AV buses.

This work aims at tackling this topic by presenting results from a real-world experimental study carried out at Tallinn University of Technology campus and involving 176 pedestrians interacting with the ISEAUTO autonomous driving shuttle. For further details about the ISEAUTO autonomous shuttle and connected research, please refer to [5–7]. We present here the results from the survey collected during the experimental sessions. The survey and interview results are evaluated via focus group interviews with key public sector representatives from the Ministry of Economic Affairs and Communications, the Estonian Transport Administration, and the City of Tallinn. Based on the findings, we propose an architecture for a Mixed Reality (MR) simulation application aimed at designing, testing, and assessing external and internal HMIs for AV shuttle interactions with pedestrians and passengers. This approach can deliver fast prototyping of different design solutions, repeatability, and an inclusive and safe testbed for interface validation with human agents. VR technologies and devices could easily integrate physiological sensors and motion-tracking systems to optimize the understanding of the street user response to AV movements and behavior, allowing for fast adaptation to different requirements and user skills and abilities.

## 2. Related Studies

The recent literature shows that the two main aspects of Human–AV interaction (HAVI) are pedestrian–vehicle interaction, which is addressed in works such as [8] or [9], and pedestrian trajectory prediction [10]. One of the first open traffic use cases of the autonomous vehicle shuttle and passenger acceptance analysis [11] in Estonia and Europe gave rise to the creation of questions for interviews conducted with people in traffic, which are referenced here. Pedestrians often use explicit means of communication, such as a handwave, to resolve conflicts in traffic scenes, e.g., yielding to the driver or requesting the right of way [1]. Pedestrian–vehicle interaction usually requires a model that can describe individual pedestrian motion under the influence of nearby pedestrians, vehicles, and traffic lights [12]. Many other studies focus on AV-to-pedestrian communication and the language of driving evaluation by means of real-world experiments or simulations. Furthermore, some comparable cross-country data with similar AV pilots are available in [13], which brought up the need to investigate how the authorities see the future of AVs in a broader context (resulting in such an interview, included as part of this current article).

Faas et al. [14] proposed a longitudinal study to assess the impact of AV external HMI (eHMI) on pedestrians crossing the street. The study proposed three scenarios based on the AV mode and intent communication, namely no communication (baseline), light-

based status communication, and status plus intention communication. The proposed experiment makes use of a monitor-based video projection of a vehicle approaching while detecting user crossing onset time, as well as a rich variety of subjective metrics by means of standardized and ad hoc tools. Results show how the eHMI supports the pedestrian in accepting the missing driver, while making him or her feel safer and more efficient in making a decision. The study points out that training and education is necessary to optimize the understanding of the signs and that learnability improves over time. Another study by the same authors [15] shows the importance of external HMI in the case of tinted windshields or distracted drivers, demonstrating the need of AV mode communication regardless of driver state.

The study by Ackermann et al. [16] employs a user-centered design approach to detect requirements for eHMI through focus group interviews and, in a second phase, proposes a video simulation-based experiment testing eHMI solutions with pedestrians. Both phases show interesting results for the definition of efficient human–AV communication. The interview sessions pinpointed some fundamental requirements, including the use of uniform symbolism or language, intuitive comprehensibility, clear reference to the pedestrian, and similarity to current communication. Experimental results compared HMI solutions differing for position, technology, coding type (textual or symbolic), and type of information provided. Results show that projection cues are preferred to LED light strips, while text messages are ranked higher than symbols. Nevertheless, the latter are not always understood and recognized. The windscreen was ranked as the most comfortable position for messages, and in general the users preferred pedestrian advice over AV status information only.

The Study by Dey et al. [17] explores the impact of the approaching vehicle appearance and behavior on the pedestrian decision-making process while crossing a street. Results show that while distance and behavior of the car play a crucial role in making a decision, vehicle design and appearance do not have much influence on pedestrians. Results from a field experimental study presented in [18] show that most pedestrians are able to manage a road crossing scenario with an autonomous vehicle without any eHMI or cues from the driver. Nevertheless, as some of the pedestrians hesitated or did not proceed in crossing the road, the authors foresee the necessity for a clear communication channel from the AV towards the other traffic agents. This is generally a common approach for inclusive design methodologies.

Several studies attempted an evaluation and assessment of design solutions based on pedestrian response using VR. De Clercq et al. [19], for instance, employed a VR immersive environment to test a combination of behaviors, eHMI, and vehicle type on pedestrian road-crossing scenarios. Results show that eHMI improves the feeling of safety for yielding vehicles while it does not influence decisions for non-yielding vehicles. The study demonstrates how text-based messages represent the most unambiguous communication method, which does not require any learning or training phase to understand the sign state change meaning. Texts are nevertheless not the most universal methods to deliver a message, as they require focused attention, they might not be readable by all users, and they might be more affected by weather conditions.

The study by Stadler et al. [20] argues that VR is a suitable tool to replace real-world tests on eHMI usability in AV–pedestrian communication. The experimental setup employed an immersive environment, accessed by Head-Mounted Display (HMD), and proposed different eHMI solutions (e.g., LED strip, symbols, arrows) for the virtual AV. Efficiency, effectiveness, and satisfaction parameters were assessed by means of qualitative and quantitative data analysis on users' reaction and decision process while crossing the street. The study validates the use of VR for assessing eHMI and underlines the advantages of rapid prototyping and user testing in a simulated environment. The same conclusions are supported by the study presented in Deb et al. [21]. The results support the use of immersive VR and real-world movements against different AV behaviors. The study assesses user experience (UX), usability, presence, and sickness symptoms and collects quantitative data

related to street crossing time and user head and body position in space. The conclusions are positive in terms of crossing time correspondence with real-world scenarios in the literature, usability, and cyber sickness. The prototype presented in [22] integrates eye tracking in a modular VR architecture for testing eHMI communication for autonomous public transportation buses. Sight analysis eventually supported the evaluation of pedestrian attention on the proposed communication systems and solutions. Rettenmeier et al. [23] address the AV-to–car driver communication in difficult traffic scenarios by means of eHMI solutions.

Other studies address the evaluation and design of Internal Human–Machine Interfaces (iHMI) for autonomous vehicles by making use, for instance, of hybrid on-road simulations for internal human–vehicle interface assessment. The system is described in [24] and proposes a wizard driver-based AV experience where the user is able to interact with the car console through tracked virtual hands in an immersive VR scenario. The goals are keeping the prototyping costs low, having a realistic experience, and improving the variety and quality of UIs and interactions between the user and AV. The study by Flohr et al. [25] addresses the user experience of shared AVs aimed at public transport. HMI assessment and prototyping is achieved by adopting an immersive video-based system architecture aimed at the evaluation of human factors during travel and the improvement of trust. The study by Morra et al. [26] proposes an immersive AV passenger simulator supported by physiological and qualitative data analysis to assess the user experience of AV state feedback and performance.

As already mentioned, the inclusiveness of frail subjects in the experience of use of shared AV systems should be one of the focal points of research in this field in the coming years. Considering these vehicles will be supporting local mobility, including daily routes covered by the elderly or children going to school, understanding the needs, type, and specificities of the LoD and HMI communication channels is crucial. A few recent studies have addressed these requirements by testing internal and external HMIs in ad hoc experimental setups with children, such as in Charisi et al. [27], or detecting preferences and the usability of AV interfaces in elderly users, for instance in the works by Morgan et al. [28], Voinescu et al. [29], and Othersen et al. [30]. The definition of a universal LoD that would be suitable in different contexts and open to easy interpretation from any user is nevertheless in its initial phase and needs efficient and reliable tools for faster design and assessment.

## 3. Language of Driving

Autonomous Vehicle technology has been introduced into our daily living experiences, and to date, the vast majority of accidents involving AVs have been as a result of humans hitting AVs [31]. This is happening at a rate higher than human agents hitting other humans. An imminent challenge, as pointed out in the study by Favaro et al. [32], is the transition period during which human road users have to interact and communicate with AVs. Vehicle manufacturers are already developing AVs fitted with virtual eyes or panel displays that contain messages intended for pedestrians, in an attempt to bridge this communication gap.

Many studies have tested and assessed different external interfaces for AV communication during simple interactions such as road crossing or traffic yielding. Nevertheless, the language of driving is quite complex and not yet defined in any formal manner. As introduced in the first SAE Edge article [33], driving is a highly complex task that requires drivers to communicate and interact with other road users to signal their intent and safely operate their vehicle. Drivers, cyclists, and pedestrians receive messages from other traffic agents, including micro-accelerations, braking, honking, eye contact, and physical gestures. Meaningful predictions and assumptions are inferred from these messages, allowing road users to understand the driving intention and on-road maneuvers. This form of communication is modulated by road conditions such as weather, time of day, and traffic congestion. Furthermore, this language has different dialects, determined by culture and geographical location. For instance, drivers in India have their own language to communicate their inten-

tions, emotions, and greetings via the car horn [34]. Given the intricacies of communicating and understanding the language of an air horn, there is much work to be done regarding the unspoken and underexplored language of driving.

It is evident that the LoD needs to be defined before developing software or interfaces supporting the interaction between road users and AVs. Once defined, the AV behavior can be verified against responsiveness to human communications. One last critical point in this novel field is the level of safety and acceptance of risk. As much as flawed human driving behavior is tolerated, this seems not to be a feasible option for an AV. The level of safety in traditional traffic interactions seems to be higher, probably because communication between agents is more understandable and humans can easily infer the point of view and messages of other human drivers. A massive upgrade in the language of driving for AVs may be the key to the acceptance of the risk.

Our vehicle currently provides several signaling symbols to pedestrians by means of a LED light panel. A blinking red cross pattern is used when the ISEAUTO shuttle detects an object that is on its way; it is intended to alert people when a dangerous situation might occur. Eventually, the signal aims to warn the pedestrians that they should not cross the street. Animated green arrows are displayed when the vehicle detects agents next to, or on, a crosswalk. The green arrows are an invitation to cross. The last symbol, vertical stripes, communicates that the AV has detected a pedestrian crossing (see Table 1). All symbols are displayed concurrently on one horizontal panel in the middle and two vertical panels on the sides.

**Table 1.** LED signaling patterns used by the AV bus to communicate with other traffic participants.

| Trigger | The vehicle is approaching a pedestrian crossing; pre-defined either by vector map or V2I communication | Objects detected by the sensors | Objects detected by the sensors |
|---|---|---|---|
| Situation | The vehicle is approaching a pedestrian crossing | The vehicle is approaching the pedestrian crossing and objects are detected on the zebra or nearby | The vehicle is driving on the road, and objects are detected close to waypoints or their moving trajectory is about to cross with the vehicle |
| Visualization |  |  |  |

## 4. Method

It is important to ensure that the users of public transport feel comfortable in using autonomous buses and that pedestrians feel safe sharing their environment with AVs. Therefore, it is important to collect feedback both from the passengers and pedestrians during the pilots. As we ran this particular pilot without the operators on board, it was especially important to determine what the subjects think about such a setting and what can be improved. For example, what do the passengers think about how the bus and its remote operator should communicate with them and how do pedestrians understand when they can safely cross the road? The answers to such questions are critical for the further development of autonomous buses.

### 4.1. Experimental Setup

The experimental setup included one self-driving shuttle minibus, driving fully autonomously on the TalTech campus road, which is semi-open for public traffic (see Figure 1). In addition to the vehicle, the setup consisted of two smart bus stops, one smart pedestrian

crossing, and one pedestrian crossing without a special traffic sign. Vehicle operation was monitored in real time by a remote operator.

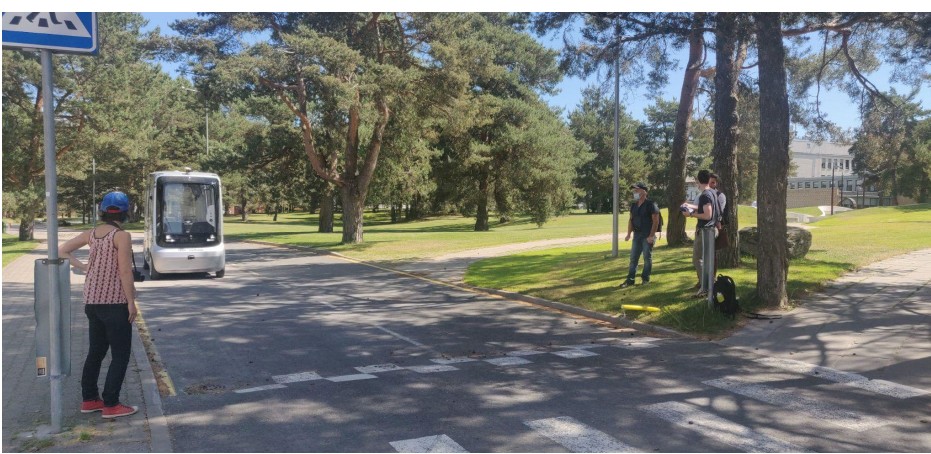

**Figure 1.** Typical pedestrian crossing scenario for AVs.

The current research focuses on the pilot projects that took place mainly during the summer months of 2021 (no winter conditions). The initial plan was also to use spring for the pilot, but due to COVID-19, we had to postpone the start. There was a pre-marketed service as a press release; it was featured on university web pages and on both the university and institute's social media, suggesting that people come and test the AV service every weekday from 4–6 PM. The time was chosen specifically because many people leave their offices at that time; thus, not just students would be interviewed, but employees. This is also the time at which local people, including the elderly, walk around the campus area.

In order to maximize the passengers' response and receive a broader opinion, we decided to conduct interviews instead of using a post-travel web survey.

The number of passengers that used the service was 539 passengers, but the answers and the analysis were based on two different aspects—those riding on an AV shuttle and those who happened to be participating in traffic scenarios with the AVs (for example, crossing the road in front of the AV shuttle). Both on-board (53 interviews) and on the side observations (176 interviews) were included.

*4.2. Focus Group Interview*

In addition to the survey, a focus group interview took place mid-February 2022 with three public sector experts on AV implementation, representing the Ministry of Economic Affairs and Communications (responsible for AV nationwide regulation), the Estonian Transport Administration (responsible for AV implementation and permits), and a transport expert from the capital City of Tallinn who was involved in the implementation of several AV pilots from the city perspective. From the research team, we had four members asking mainly pre-structured questions. The focus group interview took place online and lasted 90 min. The interview was recorded and later transcribed with Otter.AI software. The main questions relevant to this study are, for example, "*How do you envisage the AVs in Estonia, in Europe, and globally in 1, 5, and 10 years? (fully autonomous?)*" and "*How does the current infrastructure in Tallinn and Estonia allow the traffic to adapt to AVs, and what could be the changes needed in the long run?*".

## 5. Results

*5.1. Survey among Passengers*

During the pilot, we interviewed 53 passengers, who either occasionally happened to be in the area or had read about the pilot and wanted to try the AV shuttle. The pilot took place at the campus of Tallinn University of Technology; therefore, one has to take into consideration that the participants might be more knowledgeable about the existence



of autonomous vehicles as compared to the general population. Still, for many people, it was their first experience with an autonomous bus.

Of the respondents, 52.8% were men and 47.2% were women; 34% of the respondents were under the age of 18 (youth in school, aged 15–18). One quarter (24.5%) belonged to the age group 18–30 years (mostly students commuting in the campus area). Slightly over one-fifth (20.8%) belonged to the age group 31–45 years, and 17% to the age group of 46–60. Just two respondents were over 60 years of age. Just over half—52.8%—of respondents had a university degree, 30.2% had primary education (children), and 17% had secondary education (high school or vocational degree); 59.6% were employed, and the rest were students (40.4%).

The results showed that 48.1% answered that they use public transport on a daily basis, and 17.3% were weekly users; 30.8% of the respondents answered that they use it less often. Just two people (3.8%) answered that they never use any public transport. In terms of the reasons for participation, 40.4% of the respondents knew about the pilot through personal invitation to try it out, 25% had heard about it from the media, 23.1% saw the bus on the street and approached it, 19.2% had heard about it from a family member, and 28.6% received the information from their teacher or via their school/university.

First, we asked participants to rate the overall feeling of safety on a scale of 1–7. Nobody felt completely unsafe. Two people rated it as "3", and one person as "4"—mainly because of unexpected braking, and they did not have their safety belts on. Moreover, it felt unexpected that after the safety stop, the doors remained closed, and the bus audio was only in English and not in the local language. There remains much uncertainty about autonomous vehicles taking over the traffic. Many are skeptical about autonomous and non-autonomous cars driving on the streets together, which creates prejudice that autonomous cars might not be safe enough for everyday traffic. However, while riding in the AV, they saw that at low speeds, it was quite safe and there was a great focus on safety in general: 18.9% of the participants rated the overall safety with "5", and 39.6% rated it with "6". A little over one-third (35.8%) of all respondents rated the overall safety with "7". This means that over 75% of all respondents feel "very safe" while riding on the autonomous vehicle, even without the safety operator onboard. This encourages us to look further into the possibilities of connected and cooperative autonomous vehicles (CCAMs).

Next, we wanted to know how the passengers felt about their personal safety. The results were quite similar, with just minor changes: 7.6% of the passengers who gave the lowest scores ("3" or "4") and can be considered the most skeptical, explained their low grades with a worry that "at least somebody should be onboard", "the seatbelt was missing", and "it created a little phobia, being in a closed room and the robot bus communicating in a foreign language". A total of 28.3% of people rated it with a level of "5", which is substantially higher compared to the previous question, showing that people are more worried about their personal safety than about the overall traffic safety. The majority still considered it "very safe", with 30.2% of people rating it "6" and 34% rating it with "7".

Furthemore, this research aims to understand to what extent people are ready to travel by an autonomous bus. The results show that the autonomous buses could have more functions than just providing the last-mile service. While slightly less than one-third of the respondents preferred to use the service in the range of last-mile service, over half of the people answered that they would like to use the service for longer distances. However, over 60% of the respondents prefer to use the service in the range of up to 5 km. Interviews clearly showed that people feel that the vehicles are reliably controlled over such a distance. Slow speed has another strong effect on these answers, as most people felt it would be a waste of time if autonomous buses would serve longer distances. Nevertheless, almost a quarter of passengers who answered the survey (24.5%) indicated that they do not really see reasons to limit these distances. Some added that with bad weather or missing other means of transport, such a service could be really helpful, especially for the elderly and disabled people living in rural areas, where getting to the food store or to the town center can be a struggle. In addition, elderly people might even prefer a slower ride.

The survey also investigated how autonomous buses should let awaiting passengers know that the boarding has started. People who are used to public transport were sure that simply arriving at a bus stop and opening the doors is enough information for them to enter the vehicle (67.3% of all respondents). As it was a multiple-choice question, there was a significant number of respondents who wished to receive an audio signal (57.7% of all respondents). For example, this could be a voice saying that it is safe to board the vehicle. A total of 28.8% of respondents would like to have blinking lights or signs. Only one stated that there should be a button to open the doors and an audio signal that the doors are closing and the drive will start. It was rather common that people asked for audio messages in different languages, mainly because they wanted to have all possible ways of indicating information.

Going even deeper into the actual perception of a fully driverless bus service (even without the remote operator), we first and foremost wanted to know how ready people are to have such a service, where there is not even a teleoperator controlling the vehicle while they are riding in the bus. A total of 60.4% of respondents answered that they would use the service even without a remote operator. However, many people specified that they would use it if the technology is proven to be safe and it is used also by other users. A little over a quarter of respondents (28.3%) questioned the safety aspects more strongly, saying that they might use the fully autonomous shuttle services, and 11.3% answered that they would never use such a service. The biggest concern was personal safety, especially if when considering mass transit. There were respondents who added that having a remote operator who keeps an eye on the situation in the bus should be reduced to the minimum. Others feared vandalism, drunk people, or bullying and added that because of these reasons, there should always be a safety operator onboard.

We asked people whether the type of driving control of the AV, remote or autonomous, should also be communicated to the passengers; 51.9% of the respondents answered that this information is important for them, as they want to be informed at all times about what is happening with the bus, bearing in mind that this is a novel experience for everybody. Some respondents said their opinion might change in the future when AVs will be more common, and 48.1% expected public transport to be safe and not to be allowed on the streets if otherwise and thus did not need such information.

While riding on the bus, we asked people how they would want to be in contact with the remote control centre. We noticed some differences in the answers based on nationality. Estonians do not need to communicate unless necessary. People with Russian backgrounds are willing to use the remote support and wish to have more communication channels. We let people choose from several predefined options, as shown in the following figure.

As shown in Figure 2, 52.8% of respondents would like to have an onboard phone that enables them to contact the operators; 47.2% of respondents answered that just a phone number to call, when needed, is enough. The rest of the answers were substantially less popular: 17% answered that they would like to have a video call possibility with remote operators, and 15.1% did not need any kind of connection with the operator, referring to their previous public transport, where they did not communicate with drivers or the public transportation office regarding any issues. Only 7.5% of respondents wanted a continuous video and audio stream from the bus to the remote control centre. The goal of this question is twofold, as it indirectly addresses privacy issues; passengers willing to be connected to the remote operator (using an audio–video stream) are also less concerned about data privacy.

When asking the passengers when they would use such a service the most, we gave the interviewees a specific set of choices. Passengers' answers were equally distributed between using self-driving buses for daily commuting (26.4%) and in closed areas such as campuses, industrial parks, airports, hospitals, etc., or in bad weather conditions (24.5%). Surprisingly, the possibility of using the service as a link to transport hubs or other public transport options was not the most popular answer. This could have been affected by the actual area where the pilot took place, as there were no visible bus stops in the vicinity.

As people could choose only one answer, the most preferred option was using the service for daily commuting, which can include many of the other reasons mentioned separately. Some passengers who gave a different answer added that they would also use it on other occasions.

**Figure 2.** Results to the question "How would you like to be in contact with the remote operator?".

At the end of the interview, passengers were asked to rate their overall experience on a scale of 1–5. The average score was 4.44, which can be considered very high. While 50% of all participants gave "5", and 46.2% gave a strong "4", only one person gave a "3", and one person a "2".

*5.2. Survey among Other Road Users*

The second part of the survey during the pilot in Tallinn involved a different subject group in the assessment of the pedestrian-to-AV interaction in normal traffic conditions. This survey was conducted among the other participants in the traffic, who were not riding on the AV. This mostly includes pedestrians, and to a lesser extent, the users of light modes of transportation (bicycles and scooters). Visual communication between people in traffic plays an important role in dealing with different situations. For example, pedestrians rely on visual contact with drivers to assess whether it is safe to cross the road. As the end goal of building AVs is to remove safety operators from autonomous buses, it is important to collect feedback on how autonomous buses without an operator on board should and could communicate with other traffic participants (Language of Driving—LoD). To gain more in-depth insight, we gathered input through 176 structured interviews, which were conducted on the streets of the pilot area at the TalTech campus.

Out of 176 people, 50.3% fell into the age group of university students (18–30 years), 17.1% were aged 31–45, and just as many were aged 15–18. A total of 11.4% were aged 46–60. The smallest response rate was from the 60+ age group, with only 4.1% of total respondents. Of the respondents, 54.5% male and 45.5% female, and 51.3% of interviewees had a university degree (the pilot took place in TalTech campus area), 30% had secondary education (high school/vocational school), and 18.7% had primary education.

The first question aimed at assessing the state of awareness of the pedestrians towards the AV and the fact that there is no driver in the vehicle. The results show that 93.7% were aware of this, and only 6.3% did not understand that the bus was moving totally by itself.

Interestingly, 68.2% of respondents answered that they were not uneasy or intimidated by the driverless vehicle. Many of them added that the autonomous buses are so slow that, if necessary, they have plenty of time to step aside or react. Finally, 31.8% of people answered that they feel uneasy and intimidated while sharing the road with autonomous vehicles; many of them felt it is just odd and noticeable, and thus it makes them be more attentive.

On the pilot site, there was one "official" and one "unofficial" pedestrian crossing. The latter refers to a place where it is convenient for people to cross the road. This gave us a

possibility to also address the safety issues related to crossing the road in front of a fully driverless vehicle (see Figures 3 and 4).

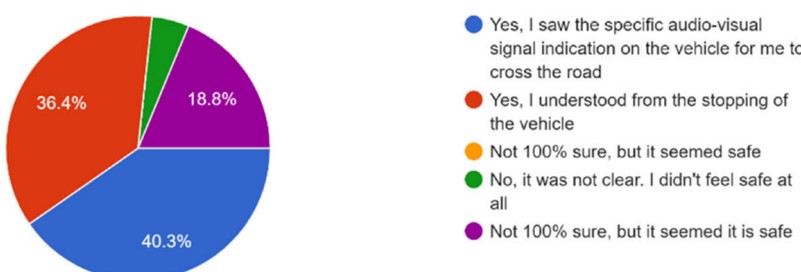

**Figure 3.** Answers to the question "Was it clear that you were able to cross safely while the vehicle had stopped at the crossing point?".

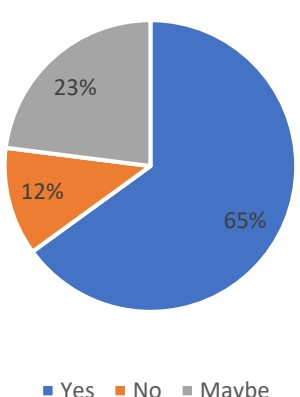

**Figure 4.** "If the vehicle would give me clear audio or visual signs to safely cross in a non-crossing area, I would feel confident to cross there too".

As shown in Figure 3, only 40.3% of the respondents answered that they saw the specific audio-visual signal indicating that it was safe to cross while the vehicle was stopped in front of the pedestrian crossing; 36.4% of the respondents answered they understood that it was safe to cross solely from the stopping of the bus; 18.8% were not sure, although they did cross the road as it seemed safe; 4.5% of the respondents answered that it was not clear at all whether they could cross the road, and they did not feel safe to do so. The results show that there is room to improve the audio-visual signals by making them more understandable. It is of utmost importance that people feel safe in such situations, as it will also affect the overall perception of safety in traffic with autonomous vehicles.

When we asked about whether the respondents would feel confident to cross the street in a non-crossing area if the bus were to give audio-visual signs that it is safe to cross, 65% of the respondents answered affirmatively, 23% answered "maybe" and added that the decision would depend on the overall street traffic, and 12% were sure they would not cross the street even if the vehicle gave them the priority and yielded. We also have to keep in mind that some people would not cross anywhere where it is not strictly permitted.

A further question addressed the topic of who should be in control of the vehicle, as there might be a physical operator or a computer operator using artificial intelligence

(AI) to drive the shuttle. This resulted in creating confusion for some of the respondents. Many of them did not understand what exactly artificial intelligence (AI) means, including its limitations and how it makes decisions. As the survey was conducted at the university campus, many of the respondents were connected to the university as students or employees and had a better understanding. Therefore, we did not have to provide a very in-depth explanation of AI to them. Most of the respondents considered AI to be more intelligent than a human, having the knowledge, precision, and even better and faster reaction time than humans do.

Of the respondents, 61.7% answered that AI should have control over the vehicle; however, in more difficult situations such as at crossings, a remote operator should take over. Over one quarter of the respondents indicated that full AI mode is also acceptable, and 12% thought that the vehicle should be under the remote operator's control at all times. This reveals a well-known aspect of technological anxiety, as passengers and other road users prefer the remote operator to take control in places where the vehicle interferes with road users.

### 5.3. Additional Suggestions from Participants

During the two surveys, we also received a number of comments as open suggestions for the future development of autonomous robot buses. Such comments covered several topics, which are summarized below.

- Speed—people want the buses to drive faster (this was the most common comment).
- Vehicle-to-passenger communication:
  - a    The bus should preferably deliver messages in several languages;
  - b    The bus should be more communicative and explain what and why it is doing something (e.g., why the bus has braked suddenly);
  - c    There was a suggestion to play lounge music inside the bus.
- Language of driving:
  - - More audio should be used to communicate with other participants in the traffic, as the signs shown in Table 1 were not fully understood or it was hard to see them under direct sunlight. The audio message should also be shown as text on the screen(s), as it can be hard to hear in traffic;
  - - Some suggested using only text instead of the signs in Table 1.
- Design:
  - - Both the interior and outer design should be more appealing;
  - - The use of brighter colors was recommended to better differentiate AV buses from regular vehicles.
- Smart bus stops and the size of the bus:
  - - The one who orders the bus should also be the one who enters the vehicle, as these shuttles are quite small in size, taking up to 6 people;
  - - The size of the bus should be bigger and accommodate at least 10 people.
- Passenger and traffic safety:
  - - Worry about not having a safety operator was expressed, as some passengers or even vandals might damage the bus;
  - - Passengers also worried about the missing seatbelts, which can easily be added and, according to the law, should be there if there is a wish for such buses to be operated in open traffic.

### 5.4. Survey among Public Sector Experts

This section summarizes the focus group interview conducted with three public sector AV experts (see also Section 4.2). On one hand, the public sector experts stated that the technology is not ready for fully autonomous service, but they envision it to be in the future. For example, the Tallinn City Transport Expert commented, *"I think accessibility*

*is a very powerful concept because if you talk about automation, then you do not need a driver anymore. You do not have to have a driver's license anymore in the future and everybody can order AVs in front of their homes".* However, no expert could predict when this would happen—either in 15 years, 50 years, or maybe never. The experts agreed that predictions made 5–10 years ago were more optimistic than the reality of today. Furthermore, this fully automated service does not work for everybody; for people with disabilities, this process likely cannot be fully automated, as there is still a need for care workers to assist people, according to the Transport Policy Expert working for the Ministry of Economic Affairs and Communications. In addition, one counterargument for fully automated transport is the social dimension—people appreciate human connection, even in the transport system.

Among the experts, the question regarding responsibility makes fully automated systems more complex. In legal practice so far, the operator (either remotely or on board) is mainly liable in the case of accidents. Even though there could be real issues could with the hardware and software of the vehicle, assigning this responsibility to companies is currently a more challenging process, and by default, the operator is responsible. However, the experts agree that the issue is more difficult when a single operator is assigned to multiple AVs. Furthermore, another issue is how to deal with remote operators that are outside the legal framework—e.g., outside Estonia/or even outside the European Union.

Technology readiness remains a key challenge on this path toward fully automated vehicles in open traffic. Operationally, AVs are not considered equal to human-driven vehicles. According to the expert working for the Estonian Transport Administration, for safety reasons, they do not allow AVs to carry out road tests over 30 km/h, which is a threshold where more critical accidents start to happen. However, in the future, automation could potentially increase safety, as most accidents are currently caused by human error, and this is expected to decrease with AVs. All experts agree that safety remains central to AVs, both as an enabler and as a key risk. The experts also agree that the potential efficiency when AVs are shared is an enabler of this technology, as this can reduce congestion, cut emissions, and make the transport sector less labor-dependent. The experts also pointed out additional risks, for instance, cybersecurity and the hybrid transport environment, including communication between normal cars and AVs. On the other side, the advancement of mobile internet connections, such as 5G and 6G, could be an enabler for this technology to move forward.

One more radical consideration stated was that the change in infrastructure is irrelevant, as AVs could cope with any infrastructure. However, the experts agreed that common international standards are important to speed up the development of infrastructure. Internet connectivity and digital standards (e.g., transport communication APIs) are the key in this regard.

*5.5. Survey Summary and Interview Conclusions*

As autonomous vehicles are novel traffic agents, people seem to be very positive about them in their reactions and their assessment. At the same time, many cannot compare them other than against the usual public transport. One of the most important outcomes from the interviews is that people are wishing and waiting for the first- and last-mile shuttles to come into operation and serve more specific destination needs, as the lengthening of the existing public transport is meant for the masses.

The interviews took place in fair weather conditions, and interviewees did not have heavy items to carry. Nevertheless the users foresee the advantages of AV in unexpected circumstances, for instance, in adverse weather. The most important factor in using AV services is safety. The passengers rated safety with high scores, at least in an area (the university campus) without particularly heavy traffic. Although the bus was driving on a public road with some overtaking of parked cars, and was being overtaken by other faster vehicles, according to the answers, the passengers felt comfortable regardless of their gender or age; however, most people do not expect AVs to be driving fast or for long distances.

The machine requires adequate means for communication, to be able to make humans feel safe in traffic and to respond to human interactions and signals. At the present, the channels supporting vehicle-to-user communication are limited to visual signage, lights, and audio signals, recorded or in real time. Moreover there are limitations in the embodiment of such visual signals in the vehicle. These signs have to be universally understood and translatable in an instant. Building many different signals on the vehicle requires time-consuming design, prototyping, and extensive testing, as in the current stage only three different signals were tested (see Table 1). These were visible, and people understood the red cross and green arrow by the colors—red always signals that something needs your attention and green is giving "the green light". The crossing zebra sign was confusing (the sign for letting people know that the shuttle is acknowledging the pedestrian crossing) as was turning the sign into green arrows once it stopped, to show that it is safe to cross.

There are other ways to communicate in the language of driving, such as the text in the front panel with short but well-understood messages; however, language might be an issue, unless it says "stop" or something universally understood. This area remains to be studied thoroughly, as it can be considered an open field of research.

## 6. Avenues for Future Research

The results drawn from the interviews and survey reveal a series of criticalities and limitations, both in the design and assessment methods of the system, which we believe can be, at least partially, overcome by adopting Extended Reality (XR) technologies and simulations. Future works intend to create modular simulation scenarios aimed at the design, prototyping, and assessment of internal and external HMIs specific to AV shuttles, facilitating the evaluation of users' behavior and response. Some of the most problematic aspects of the passenger experience in the ISEAUTO autonomous bus are related to system modality feedback, teleoperated or in autonomous mode, emergency state, door opening signals, AV positioning and route navigation, and the type of communication and support that would be feasible to offer from a remote operator in case of necessity. Another important aspect is the inclusiveness of the messages, which in the experimental study only referred to language and cultural background differences. It is also clear that the presented experimental setup and methods have some flaws related to users, repeatability, and adjustability of the machine parameters and features. With the proposed scenarios, we intend to create a XR testbed that would enable fast and safe assessment prior to real world investigations.

The first proposed scenario focuses on Internal Human–Machine Interfaces (iHMIs), which would support wellbeing, a feeling of safety, and efficient communication on the state of the system from the AV to the passengers. This specific topic is not yet properly addressed in the literature, but it is essential for the future development and success of sustainable local mobility based on autonomous buses. The test of iHMIs for the ISEAUTO shuttle can be achieved by contextual experiments supported by Augmented Reality (AR) hardware solutions (e.g., see-through head mounted displays). By overlapping ad hoc interfaces on existing objects and augmenting real-world scenarios, we aim at testing multiple iHMI alternatives, such as navigation dashboards, audio visual alarms, video and voice operator feedback, holograms, and buttons or touch displays, without any physical modification to the original AV structure and body.

The second scenario will focus on testing and prototyping different solutions for the AV external HMI (eHMI), including the language of driving (LoD). An immersive virtual scenario would support the validation of existing solutions in a safe and repeatable experimental setup, which could be more inclusive for frail subjects (impaired users, elderly people, children). This would also allow the fast prototyping of new interfaces by rapidly modifying and assessing their position, size, the type of messages, language, and communication channels (voice, sound, gestures, etc.) [34]. The simulator would also support the test of user–AV interaction in specific road scenarios, including emergency

situations, collisions, road crossings, and bus stop interactions (e.g., door opening speed). Moreover, the integration of simultaneous agents' interactions and the detection of user response to different vehicle types or design solutions for the ISEAUTO shuttle would be easily achieved in a virtual environment. The development of simulation-based user-centered studies enables the easier integration of technologies aimed at physiological data collection and analysis. We believe that passengers and other road users' stress levels can be assessed by detecting heart rates and analyzing heart rate variability during the experimental sessions. Passengers' attention can be studied by collecting eye and head movement data, in relation to the size and position of the proposed interface solutions. Conclusions can also be inferred from the analysis of posture and movements while, for instance, the user is virtually approaching a crossroad or the AV at the bus stop. The flexibility of XR scenarios allows the fast setup and assessment of specific use case tasks that the passenger might have to perform (contacting the operator, unlocking the doors, interacting with a dashboard, purchasing a ticket).

The third proposed scenario focuses on operator and XR Digital Twin (DT) tools for remote maintenance and servicing. DTs are simulations of complex systems and processes, which are updated and synchronized through the exchange of data and information with the real-world counterpart. DTs are already successfully used in industry for control and teleoperation [35]. This is a fundamental aspect both from a technical point of view and from the point of view of the users' feelings of trust, safety, and support while using the AV. The AV shuttle ISEAUTO propulsion engine DT is already used for performance prediction and prognosis, as presented in [36]. ISEAUTO is proposed as an integral unit of the Industry 4.0 environment [37]. Extending the synchronized digitalization to other AV systems and integrating them in an XR interface would support the operator in assessing the state of the shuttle with a more precise visualization and facilitate the decision-making process. Advanced interaction methods enable the operator to interact virtually with the real AV shuttle while in a remote location. Using the same simulation in AR would support the potential local operator with remote guidance and instructions.

## 7. Conclusions

This paper studied the new challenges of the emerging topic of the so-called language of driving (LoD) in connection to transportation systems in smart cities. The study was conducted through field interviews at Tallinn University of Technology campus using our custom ISEAUTO autonomous driving shuttle. This research integrates interviews involving 53 riders and 176 road users interacting with the AV shuttle as well as an expert focus group to build a joint base of needs and requirements for AVs in public spaces. According to the questionnaire results, people feel safe onboard but are still interested in knowing whether the bus is remotely operated or fully automated. Furthermore, the signaling (communication) from the bus to other road users was perceived as comprehensible, though with some limitations involving the position of the signals on the bus and the specific type of signals shown. Signal appearance, function, and position can be developed more efficiently using VR technology before adoptions in AVs.

The adoption of an XR-based experimental setup offers many advantages, which can support the faster definition of an inclusive and standard LoD for AVs. The adoption of advanced visualization and interaction technologies grants extended testing on a wider range of users and in a larger number of use cases. Simulated scenarios allow the easier assessment of usability and acceptance of the system by providing valuable solutions for real-world user experiments, criticalities, and issues.

Future works will integrate the available hardware technologies in a multi-scenario application developed for AR/VR testing and assessment, while recruiting users of different ages and with diverse abilities.

**Author Contributions:** Conceptualization, K.K., S.-L.P., R.-M.S., R.S. and M.B.; methodology, K.K., S.-L.P., R.-M.S. and R.S.; software, K.K., S.-L.P. and M.B.; validation, R.-M.S., R.S. and M.B.; formal analysis, K.K. and S.-L.P.; investigation, K.K. and S.-L.P.; resources, K.K., S.-L.P. and R.-M.S.; data curation, K.K., S.-L.P. and M.B.; writing—original draft preparation, K.K., S.-L.P., R.-M.S. and R.S.; writing—review and editing, M.B., K.K. and S.-L.P.; visualization, K.K. and M.B.; supervision, R.-M.S., R.S. and M.B.; project administration, K.K. and M.B.; funding acquisition, K.K., S.-L.P. and R.S. All authors have read and agreed to the published version of the manuscript.

**Funding:** This work was supported via funding by two grants: the European Union's Horizon 2020 Research and Innovation Programme, grant agreement No. 856602, and the European Regional Development Fund, co-funded by the Estonian Ministry of Education and Research, grant No. 535 2014-2020.4.01.20-0289, and was conducted using the Smart Industry Centre (SmartIC) core facility funded by the Estonian Research Council, grant TT2.

**Institutional Review Board Statement:** Ethical review and approval were waived for this study due to all humans participating had to participate also in face to face interviews and prior to that they were informed, that they participate in scientific research, carry their own risks and will stay anonymous. No animals were involved.

**Informed Consent Statement:** Informed consent was obtained from all subjects involved in the study.

**Data Availability Statement:** The data presented in this study are available on request from the corresponding author. The data are not publicly available due to that it hasn't been made public before this article.

**Acknowledgments:** The financial support from the Estonian Ministry of Education and Research, Estonian Research Council, and the Horizon 2020 Research and Innovation Programme is gratefully acknowledged.

**Conflicts of Interest:** The authors declare no conflict of interest. The funders had no role in the design of the study; in the collection, analyses, or interpretation of data; in the writing of the manuscript, or in the decision to publish the results.

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
