# Peer review of "Language of Driving for Autonomous Vehicles"

_applsci, doi:10.3390/app12115406_

Round 1
Reviewer 1 Report
This is an interesting paper. The study analyzes the perception and predisposition to use connected driverless vehicles as passengers. The research topic presents important practical implications, considering the growth of technology in the field of traffic and the benefits they provide for this type of vehicles. From my perspective, the changes needed are considerably achievable through a phase of author revisions.
The introduction adequately explains the study variables, and it offers a quite complete background. However, more emphasis could be placed on the evidence assessing the acceptance of autonomous vehicles and other technological systems in the field of traffic. In this sense, there is evidence showing the reasons behind the use/lack of use of this type of vehicles. Thus, it is important to highlight the concerns of users on data privacy, technology-related anxiety, perceived safety and hedonic motivation as moderating variables in the intention of use (e.g. https://doi.org/10.3390/app12010103 and https://doi.org/10.1016/j.trc.2020.102732). Further, I am not sure if these issues can be considered as adequately validated in the discussion of the paper. Therefore, please raise this point (at least briefly) while discussing the outcomes of the study.
Also related to the discussion, the current way of raising it is sometimes difficult to follow. I recommend using a discussion that explains the obtained results and contrasts them with other similar research. You must include aspects related to attributions that users give to autonomous vehicles. This way, the perceived safety, perceived risk and privacy of personal data are essential to understand the attitudes and behaviors of users. Therefore, the intention of use and the perceptions of users for what concerns autonomous vehicles will be determined by these variables (e.g. https://doi.org/10.1016/j.ssci.2019.07.041, https://doi.org/10.1016/j.tra.2018.10.019 and https://doi.org/10.3390/app12094759). Given this relevant role and your survey-based research design, a good -and sufficient- approach and discussion about user-related (human) factors may explicitly support the practical points raised by your study.
Author Response
Reviewer 1:
Answer: Thank you for your valuable time in reviewing this paper. Your suggestions have been used to improve this paper and will be used in the future to improve the overall quality of our work.
Comment 1: More emphasis could be placed on the evidence assessing the acceptance of autonomous vehicles and other technological systems in the field of traffic. In this sense, there is evidence showing the reasons behind the use/lack of use of this type of vehicles. Thus, it is important to highlight the concerns of users on data privacy, technology-related anxiety, perceived safety and hedonic motivation as moderating variables in the intention of use (e.g. https://doi.org/10.3390/app12010103 and https://doi.org/10.1016/j.trc.2020.102732). Further, I am not sure if these issues can be considered as adequately validated in the discussion of the paper. Therefore, please raise this point (at least briefly) while discussing the outcomes of the study.
Reply: Thank you for your valuable comments. We do agree with your concern, and in the view of the authors it is important to bring out this evidence. Part of this information is also gathered in our interviews, specifically in section 5.1, in which technological anxiety in terms of “worry” about the gaps in this technology. To better highlight this issue, we have added the following text in section 5.1 line 412-414
“The goal of this question is twofold as it indirectly addresses the privacy issues, passengers willing to being connected to the remote operator (using audio video stream) are also less concerned about data privacy.”
and section 5.2 lines 504 – 506.
“This reveals a well-known aspect of technological anxiety as passengers, and other road users, prefer the remote operator to take control in places where the vehicle interferes with road users. “
There is no gender differentiation as this falls out of the scope of the study, in the opinion of the authors the autonomous vehicles should be safe and operational for everybody regardless of the gender and ethnic groups.
Comment 2: Also related to the discussion, the current way of raising it is sometimes difficult to follow. I recommend using a discussion that explains the obtained results and contrasts them with other similar research. You must include aspects related to attributions that users give to autonomous vehicles. This way, the perceived safety, perceived risk and privacy of personal data are essential to understand the attitudes and behaviors of users. Therefore, the intention of use and the perceptions of users for what concerns autonomous vehicles will be determined by these variables (e.g. https://doi.org/10.1016/j.ssci.2019.07.041, https://doi.org/10.1016/j.tra.2018.10.019 and https://doi.org/10.3390/app12094759). Given this relevant role and your survey-based research design, a good -and sufficient- approach and discussion about user-related (human) factors may explicitly support the practical points raised by your study.
Reply: The scope of the paper is to describe the language between humans and machines and the issues that it raises in today’s advancements of the technology and the concern of people. The perceived safety is addressed in several points of Section 5.1 and Section 5.2. Furthermore, we truly think that addressing the attitude and behaviour of people in connection to AVs is essential for the development of such technology. This aspect will be further investigated in the prosecution of our research. One example is mentioned in the paper by using VR and AR tools to conduct these pilots to be more efficient, before taking the pilots again to real environment. Thus, these suggestions require further investigation.
Reviewer 2 Report
Please see attached file

Author Response
Reviewer 2:
Comment 1: However, for the fifth session of “Results”, most of the findings reported are relevant to “autonomous vehicles acceptance analysis” with qualitative description, rather than directly addresses the issue of language of driving. It seems a bit inconsistent with the article title and the setting of the first three sections.
Reply: These surveys were conducted as face-to-face interviews in the autonomous vehicles and on the sidewalks of our pilot area. The topic of the language of driving is addressed in Section 5.2, where for example the authors state:
„Visual communication between people in traffic plays an important role in dealing with different situations. For example, pedestrians rely on visual contact with the drivers to assess whether it is safe to cross the road.“
It is important to emphasize that, charts 2 and 3 are discussing the visual-audio communication between the vehicles and the road-users.
Comment 2: My main point for improvement: How would the author(s) assess the contributions or limitations of this research, compared to other “Language of driving” studies?
Reply: We have used only a literature comparison to other studies in Section 2, but our goal was to see it ourselves in traffic cultural space - how do people address the issues in real time traffic, when they have to make decisions sharing the roads with fully autonomous vehicles. In this type of research is very hard to compare to other study in the results section as the variability of the sample and questions makes the outcome not-comparable.
Our paper was not taking the focus of addressing the issue of comparing with the other studies, it was to how we can utilize AR/VR and what are the real issues that come out in traffic for example by providing a common framework for others to compare.